# Cosmological Test of an Ultraviolet Origin of Dark Energy

Hans Christiansen, Bence Takács  and Steen H. Hansen *

Dark Cosmology Center, Niels Bohr Institute, Jagtvej 155, 2100 Copenhagen, Denmark;
hans.christiansen@nbi.ku.dk (H.C.); bktakacs@gmail.com (B.T.)
* Correspondence: hansen@nbi.ku.dk

**Abstract:** The accelerated expansion of the Universe is impressively well described by a cosmological constant. However, the observed value of the cosmological constant is much smaller than expected based on quantum field theories. Recent efforts to achieve consistency in these theories have proposed a relationship between Dark Energy and the most compact objects, such as black holes (BHs). However, experimental tests are very challenging to devise and perform. In this article, we present a testable model with no cosmological constant in which the accelerated expansion can be driven by black holes. The model couples the expansion of the Universe (the Friedmann equation) with the mass function of cosmological halos (using the Press–Schechter formalism). Through the observed link between halo masses and BH masses, one thus gets a coupling between the expansion rate of the Universe and the BHs. We compare the predictions of this simple BH model with SN1a data and find poor agreement with observations. Our method is sufficiently general to allow us to also test a fundamentally different model, also without a cosmological constant, where the accelerated expansion is driven by a new force proportional to the internal velocity dispersion of galaxies. Surprisingly enough, this model cannot be excluded using the SN1a data.

**Keywords:** dark energy; black hole physics; large-scale structure of the Universe; galaxies: kinematics and dynamics



## 1. Introduction

The accelerated expansion of the Universe was originally observed in SN1a data [1,2]. Subsequently, these findings have been confirmed by a range of independent observations, including the growth of large-scale structures and the cosmic microwave background [3–8], all of which indicate that a cosmological constant, represented by $\Lambda$, apparently provides excellent agreement with all observables. This is quite remarkable because it implies that the cosmological standard model fits nearly all astronomical observations with just a handful of free parameters, one of which is the energy density represented by the cosmological constant $\Lambda \approx 3 \times 10^{-122} l_P^{-2}$, where $l_P$ is the Planck length.

However, a significant problem arises, as a quantum field explanation of the magnitude of $\Lambda$ is off by approximately 120 orders of magnitude [9,10]. This discrepancy has led theoretical physicists to contemplate: "If a solution to the cosmological constant exists, it may involve some complicated interplay between infrared and ultraviolet effects (maybe in the context of quantum gravity)" [11].

The concept of linking the largest scales (cosmological constant on cosmological scales) with the most compact objects (such as black holes) was explored by Cohen et al. [12]. They discussed effective field theories with a cut-off scale $\lambda$, where the entropy in a box of volume $L^3$ is $S \sim L^3 \lambda^3$. However, the Beckenstein entropy [13,14] of a black hole has a maximum value of $S_{Be} \sim L^2$. This discrepancy may lead to inconsistencies when dealing with very large objects like the entire Universe. To address this issue, Cohen et al. [12] proposed a relationship between the UV cut-off and the IR physics to ensure that effective field theories remain consistent. This idea has garnered significant interest in the theoretical physics community over the last few years [15–17].

One crucial missing element between the observation of the accelerated expansion of the Universe and the range of theories suggesting a connection between the IR and UV phenomena is a testable model. In this article, we present a phenomenological model that contains no cosmological constant. Instead, the model calculates the time-dependence of the Universe's expansion based on the evolution of the abundance of large-scale structures. Cosmological structure formation follows a bottom-up process, where small structures merge to form larger structures, resulting in a time-evolution of this new effect.

Since it is uncertain whether the UV-IR connection should be fundamentally linked to the entropy of black holes raised to some power [12,15], the velocity dispersion of dark matter in cosmological halos [18,19], or something entirely different, we introduce a single parameter, denoted as $\beta$, along with a normalization, to encompass all these cases. This way, we introduce a new "force" that is proportional to the sum of $\sum M_{halo}^{\beta}$, where $M_{halo}$ represents the mass of cosmological halos. By comparing the resulting cosmological expansion with SN1a data, we find that this phenomenological model, without a cosmological constant, provides a temporal evolution that appears to be approximately as good as the standard $\Lambda$CDM model.

## 2. The Basic Idea

The expansion of the Universe is independent of the number of halos in the standard description of cosmology. This is seen by the fact that the Friedmann equation, which describes the expansion of the Universe, can be written as

$$\left(\frac{H}{H_0}\right)^2 = \Omega_{M,0}\, a^{-3} + \Omega_{\Lambda,0}\,, \tag{1}$$

where the Hubble parameter is given by $H = \dot{a}/a$, $a(t)$ is the radius of the Universe normalized to unity today, and all quantities with sub-0 represent quantities today, such as $\Omega_{M,0} = 0.3$ and $\Omega_{\Lambda,0} = 0.7$. This equation may be described by $a(t) = a(\Omega_M, \Omega_\Lambda)$, and one can include terms for radiation and curvature in the equation as well.

Knowing the expansion history of the Universe, one can now calculate the number of halos of a given mass as a function of time $N(M, t)$. One example of this is given by the Press–Schechter formalism [20], which will be discussed in detail below. Using the fact that the expansion is a function of time $a(t)$, the distribution of halos can be described by $N(M, a(t))$.

Instead, as will be shown below, by introducing a new energy term related to the distribution of halos, one can get a new Friedmann equation, which looks like

$$\left(\frac{H}{H_0}\right)^2 = \Omega_{M,0}\, a^{-3}(1 + F[N(M, a)])\,, \tag{2}$$

with no cosmological constant. The function $F[N]$ depends on the distribution of masses of cosmological halos. As a concrete example, one can use the observed connection between the halo masses and the black hole masses (extrapolated to be valid at all masses); one thus sees that the expansion may be written as a function of the distribution of BH masses. The change from the standard Friedmann equation to this model can, hence, be described by

$$a(\Omega_M, \Omega_\Lambda) \rightarrow a(\Omega_M, N(M, a))\,. \tag{3}$$

It is important to clarify the following point. Observational data, such as that from CMB and SN1a, show that Equation (1) provides an excellent fit with an essentially flat Universe. If we instead calculate the expansion of a Universe using Equation (2), then one may get an accelerated expansion very similar to that of the $\Lambda$CDM model. This implies that if we were to analyze the corresponding data in that Universe from CMB or SN1a with Equation (1), then we would again conclude that the Universe is flat. A detailed discussion on this point was made by Linder and Jenkins [21], who wrote the corresponding RHS of

our Equation (2) as $\Omega_M a^{-3} + \delta H^2 / H_0^2$, and they wrote: "all we have observed for sure is a certain energy density due to matter, $\Omega_m$, and consequences of the expansion rate $H(z)$".

## 3. The Press–Schechter Formalism

The evolution of the number of structures of mass $M$ as a function of cosmic time was first derived by Press and Schechter [20]. Under the assumption that primordial density perturbations are Gaussian, the distribution of the amplitudes of perturbations of mass $M$ will take the form

$$p(\delta) = \frac{1}{2\sqrt{\pi}\sigma(M)} \exp\left[-\frac{\delta^2}{2\sigma^2(M)}\right], \tag{4}$$

where the density contrast of a perturbation of mass $M$ is defined as $\delta = \frac{\delta\rho}{\rho}$, and $\sigma(M)$ is the variance. Such a distribution will have its variance equal to the mean of the square of density fluctuations $\sigma^2(M) = \langle \delta^2 \rangle$. Press and Schechter assumed that upon reaching some critical amplitude $\delta_c$, density perturbations will rapidly form into bound objects.

The variance of density perturbations $\sigma^2(M)$ is directly related to the mass $M$ of bound density perturbations and to the power spectrum of density perturbations $P(k)$ by

$$\sigma^2(M) \propto A M^{-(n+3)/3}, \tag{5}$$

where $n$ is the spectral index. Throughout this paper, we assume that $n \approx -2.5$, as observed at galaxy scales today [22,23]. The fraction $F(M)$ of fluctuations of masses within the range $M$ to $M + dM$, which become bound at epoch $t_c$ for amplitudes $\delta > \delta_c$, is

$$F(M) = \frac{1}{\sqrt{2\pi}\sigma(M)} \int_{\delta_c}^{\infty} \exp\left[-\frac{\delta^2}{2\sigma^2(M)}\right] d\delta, \tag{6}$$

where $t_c = \delta_c / \sqrt{2}\sigma(M)$ is the critical time. The critical time $t_c$ is related to the mass distribution $M$ by the relation (5) and is rewritten

$$t_c = \frac{\delta_c}{\sqrt{2}\sigma(M)} = \left(\frac{M}{M^*}\right)^{(3+n)/6}, \tag{7}$$

where $M^* = (2A/\delta_c^2)^{3/(3+n)}$ is a reference mass wherein information on the cosmic epoch is contained. The fluctuations evolve according to $\ddot{\delta} + 2H\dot{\delta} = 4\pi G\rho\delta$, and from [24,25], it is known that in homogeneous and isotropic cosmologies, the amplitudes of density perturbations grow according to

$$\delta(a) \propto \frac{\dot{a}}{a} \int_0^a \frac{da'}{(\dot{a}')^3}. \tag{8}$$

This equation is valid even though the expansion history is not given by a $\Lambda$CDM model; however, as we will find that the expansion history is surprisingly close to that of $\Lambda$CDM, the evolution of $\delta(a)$ will be very close to that in a $\Lambda$CDM Universe. We will, nevertheless, solve this equation numerically as a function of the actual expansion history of our model.

We can now incorporate time implicitly into $M^*$ as

$$M^* = M_0^* \left(\frac{\delta(a)}{\delta(a_0)}\right)^{6/(3+n)}. \tag{9}$$

By assuming that $M = \bar{\rho}V$, $\bar{\rho}$ is the mean density of the background, and $V$ is the volume, one obtains

$$N(M) = \frac{\bar{\rho}}{\sqrt{\pi}} \frac{\gamma}{M^2} \left(\frac{M}{M^*}\right)^{\gamma/2} \exp\left[-\left(\frac{M}{M^*}\right)^{\gamma}\right], \tag{10}$$

where $\gamma = 1 + \frac{n}{3}$. The above derivation is standard and can be found in many text-books [26].

With this expression, we can now calculate the expectation value of a power of $M$ as

$$\langle M^\alpha \rangle = \int_0^\infty N(M) M^\alpha \mathrm{d}M, \tag{11}$$

and if we have ratios of such expectation values, the normalizations cancel:

$$\frac{\left\langle M_i^{\beta+1} \right\rangle}{\langle M_i \rangle} = \frac{\int_0^\infty M^{\beta+\gamma/2-1} \exp[-(M/M^*)^\gamma]\mathrm{d}M}{\int_0^\infty M^{\gamma/2-1} \exp[-(M/M^*)^\gamma]\mathrm{d}M} = M^{*\beta} I, \tag{12}$$

and the integrals can be expressed through Gamma functions.

### 4. The Revised Friedmann Equation

If one considers a new force proportional to the squared velocity dispersion of the dark matter particles in a cosmological halo, then this leads to an extra term in the Friedmann equation [18]

$$\left(\frac{H}{H_0}\right)^2 = \Omega_{\mathrm{M},0} a^{-3} \left[1 + \eta \sum \left(\frac{\sigma_i}{c}\right)^2\right], \tag{13}$$

and if one instead considers the change in energy to arise from a more general term

$$\Delta E = -\kappa \frac{Gm}{r} \sum_i M_i^{\beta+1}, \tag{14}$$

one gets a new Friedmann equation of the form

$$\left(\frac{H}{H_0}\right)^2 = \Omega_{\mathrm{M},0} a^{-3} \left(1 + \kappa \frac{\left\langle M_i^{\beta+1} \right\rangle}{\langle M_i \rangle}\right). \tag{15}$$

Defining the constant $\mu = \kappa I M_0^{*\beta} \delta(a_0)^{-2/\gamma}$ and using Equation (12), one obtains

$$\left(\frac{H}{H_0}\right)^2 = \Omega_{\mathrm{M},0} a^{-3}(1 + \mu f), \tag{16}$$

where

$$f = \left(\frac{\dot{a}}{a} \int_0^a \frac{\mathrm{d}a'}{(\dot{a}')^3}\right)^{1/p}, \tag{17}$$

and $p = \gamma/2\beta$.

The effect of the terms on the RHS of Equation (16) can be analyzed just like in the standard cosmology, where each term can be described by an equation of state with properties $\rho_j = a^{-3(1+\omega_j)}$. Thus, the first term (which is just the CDM) leads to $\omega_m = 0$, and the second term may lead to $\omega_f = -1$ in the case that the choice of $\mu$ and $\beta$ happens to lead to an expansion history similar to that of the $\Lambda$CDM Universe, as we will show below may happen for carefully chosen values.

Whereas the entire RHS of Equation (16) may be viewed as resulting from CDM, then the difference in the temporal evolution of the two terms is crucial: it has been observed that the Universe transitions from a positive deceleration parameter $q = -\ddot{a}a/\dot{a}^2$ to a negative one [27]. In Equation (16), the corresponding early effect is driven by the first term (the standard CDM term), and the transition to the late accelerated evolution is driven by the second $\mu f$-term.

We now have the new Friedmann Equation (16), which must be solved numerically. Since this model can mimic the accelerated expansion of the Universe through the evolution of all the overdensities, we will below refer to this model as a $\delta$CDM model. There are, in principle, two free parameters: namely, $\beta$, which should come from some fundamental principle (as described in the introduction), and $\mu$, which is merely a normalization of this effect. Equation (16) contains the overdensities $\delta$ on the RHS and is thus significantly more complex than the Friedmann equation of the $\Lambda$CDM model. For the numerical solution, we use a backwards differentiation formula, which is an implicit method of numerical integration suited to stiff problems.

## 5. Supernova Data

In order to test the model, we compare it with SN1a data at the apparent magnitude. In this work, we employ SN1a data from the Pantheon+ analysis [28], which includes 1550 SN1a of redshifts up to $z \sim 2$. We also calculate a simple $\chi^2$ to estimate the quality of the models, as compared to the standard $\Lambda$CDM model, and leave a proper analysis including the covariance matrix, allowing $\Omega_M$ or the spectral index $n$ to be scale- or time-dependent, etc., to the future.

In Figure 1, we present the SN1a data together with the standard $\Lambda$CDM model ($\Omega_{M,0} = 0.33$, $\Omega_{\Lambda,0} = 0.67$), and a $\delta$CDM model with $\beta = 0.17$ (and the best-fit normalization $\mu$). It is clear that the two models approximately follow the SN1a equally well. The $\chi^2$ of the $\delta$CDM is slightly bigger than that of the $\Lambda$CDM model. In the lower panel, we show the residual from the $\Lambda$CDM model.

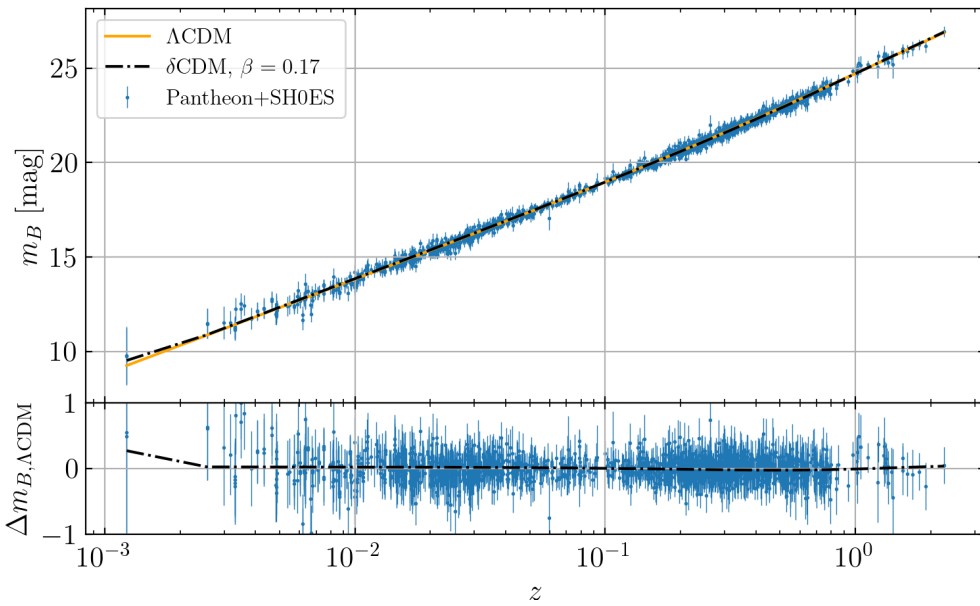

**Figure 1.** Apparent magnitudes of SN1a from the Pantheon+ analysis [28] overlaid with the apparent magnitudes of a flat-$\Lambda$CDM model and a $\delta$CDM model with $\beta = 0.17$. The residuals from the $\Lambda$CDM model are shown below. The alternative model is seen to follow the expansion history of the Universe in fairly good agreement with the SN1a data. The small kink at $z \approx 10^{-3}$ is due to the finite steps used in $z$.

In Figure 2, we present a parameter scan over a wide range of $\beta$ values from $\beta = 0.1$ to 1, and we vary the normalization parameter $\mu$. We select a range of $\chi^2$ values in fair agreement with the data (within 15% of the best $\chi^2$ of the $\Lambda$CDM model). All model parameters outside this range are color-coded white. We note a few areas of interest, all with values in the range between $\beta = 0.1$ and 0.3. The point with a black square at $\beta = 0.17$ is the model from Figure 1.

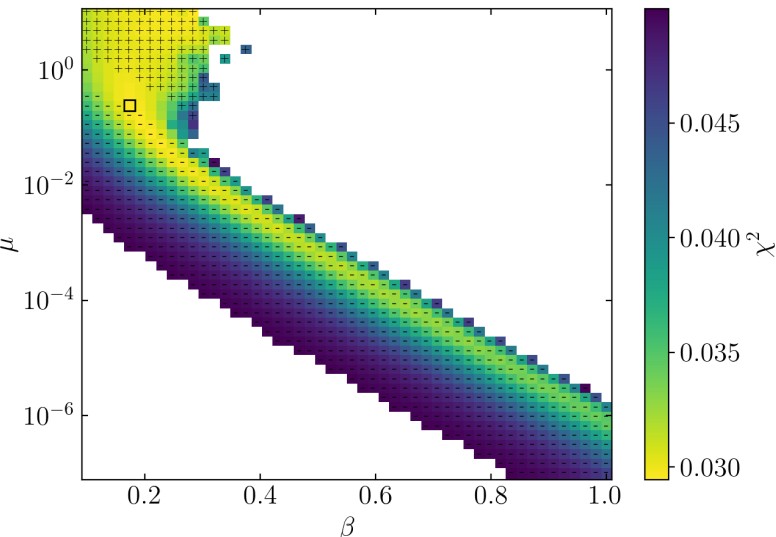

**Figure 2.** Agrid in $\beta$ and $\mu$. The color indicates $\chi^2$ calculated over the SN1a data. All points with $\chi^2$ significantly worse than that of $\Lambda$CDM have been left white. There are several points in the parameter space that have models that fit the SN data approximately as well as $\Lambda$CDM, including points in the range $\beta = 0.15 - 0.4$. All points with absolute magnitudes $M > -19.0$ are indicated with a plus sign, and a minus sign indicates $M < -19.5$. The black square at $\beta = 0.17$ represents the model of Figure 1.

We are leaving the SN1a absolute magnitude as a free parameter in the analysis. There is a clear valley at small values of $\mu$ that covers $\beta$ from approximately 0.1 to 0.6. Interestingly, some of the models have a slightly different evolution of the expansion from the standard $\Lambda$CDM model, both at high and low redshift, and we expect to quantify to what degree these models can be rejected with other astronomical observations in a future paper. Besides this valley, there are also a few models at higher $\mu$ values that fit the SN data fairly well; however, all have absolute magnitudes significantly different from the result from the $\Lambda$CDM model of $-19.3$ and from observations [29,30]. These models are indicated in Figure 2 with a plus sign if $M > -19.0$ (high-$\mu$ region) and a minus sign if $M < -19.5$ (below the valley). If we instead were to use a value of $n = -2.3$, we would find a best-fit value of $\beta = 0.27$ with parameters in fair agreement with the SN1a data within the range $0.2 < \beta < 0.4$. The most extreme (and most likely physically non-relevant) possibility is the one of the undeveloped initial spectrum of $n = +1$, which leads to a surprisingly good fit to the expansion history, with a fitted value of $\beta = 1.4$ (with reasonable values in the range $1.2 < \beta < 2.0$).

## 6. Discussion

Our phenomenological description covers a wide range of underlying models through the free parameter $\beta$. One concrete example is the assumption that the changed energy is proportional to the surface area of the black holes, and thus, that the accelerated expansion is driven by the growing black holes. It has been observed that there is a power-law relation between the BH mass and the velocity dispersion in the halo [31,32]

$$M_{\mathrm{BH}} \sim \sigma_{\mathrm{halo}}^{5.1}. \tag{18}$$

Even though this relation is best established in the range $10^6 M_{\odot} < M < 10^{10} M_{\odot}$, here we extrapolate this relation to all masses. We are thus not addressing the physical mechanism establishing the connection between the galaxy masses and the BH masses (which may be energy feedback from supermassive BHs during the galaxy formation process), but we are instead merely taking this as an observational fact. Also, the significantly large number of stellar-sized BHs should change the details in the connection between

the BH and galaxy masses beyond Equation (18). In principle, one could improve on this simplification; however, we will not attempt this here.

Observations show that halo mass and velocity dispersion are approximately connected through [33]

$$\frac{M_{\text{halo}}}{10^{12} M_\odot} \approx \left( \frac{\sigma_{\text{halo}}}{100 \text{km}/\text{sec}} \right)^3. \tag{19}$$

Since a BH's area is proportional to the BH's mass squared, we thus get $\beta \approx 2.4$. If, instead, the relevant parameter is proportional to the BH's area to the power 3/4 [12,15], then one should expect $\beta \approx 1.55$. From our analysis, we instead find $\beta \approx 0.2$, which is significantly smaller than the BH prediction. One should keep in mind that there are large uncertainties here: the connection between BH and halo masses has a spread, and also the connection between the halo mass and velocity dispersion has a non-trivial spread.

Another model suggests a connection between the accelerated expansion of the Universe and the velocity dispersion of dark matter in cosmological halos [18,19], which predicts $\beta = 0.5$. Since the mass function can be described with a scale-dependent power spectrum with a spectral index going from approximately $n \approx -2.5$ at the smallest scales to $n \approx -1$ at galaxy cluster scales [22,23], the true evolution is found by integrating over the full mass distribution rather than simplifying with a single spectral index as we have done here. We note that using a spectral index around $n = -2$ would lead to an accelerated expansion of the Universe in fair agreement with SN1a data using $\beta = 0.5$, and we therefore conclude that the present analysis cannot exclude the suggestion of refs. [18,19].

We have seen above that with an appropriate choice of the free parameter $\beta$, one can get an expansion history of the $\delta$CDM model in fair agreement with that predicted in the standard $\Lambda$CDM model. This implies that all the observations mentioned in the introduction, including the CMB observations, the growth of large scale structure, integrated Sachs–Wolfe effect, etc., are in agreement with predictions in this model. For instance, if the CMB data are analyzed with a $\Lambda CDM$ model, then the result is that $\Omega_M \approx 0.3$ and $\Omega_\Lambda \approx 0.7$, and if the CMB instead is analyzed with our model, then it will show that $\Omega_M \approx 0.3$ and that the accelerated expansion of the Universe results from $\beta \approx 0.17$. The fact that the expansion history of the Universe in the standard $\Lambda CDM$ model to first approximation is indistinguishable from that of the $\delta CDM$ model with $\beta = 0.17$ also implies that the halo mass function is essentially identical in these two models. A related discussion on the growth of perturbations (in the linear regime) was made by [21] using different description of the general expansion history (see also [34]).

The main point of this paper is to demonstrate that one can get an expansion history that is in fair agreement with that of the $\Lambda CDM$ model entirely without using a cosmological constant. This is exemplified by plotting the full apparent magnitude in Figure 1. Indeed, the new model first has deceleration at high redshift (when there is very little substructure), which then transitions to accelerated expansion in the later Universe, just like the $\Lambda CDM$ model. Naturally, one should expect some level of variation between the $\delta$CDM and $\Lambda$CDM models, and it will be interesting in the future to investigate if such differences may support the observational indications that possibly not even dynamic versions of the cosmological constant provide a self-consistent explanation of all the available cosmological data [35–37].

A recent study of the evolution of BH masses has also suggested a link between the BH mass increase and the expansion of the Universe [38]. That paper considered nonstandard, singularity-free BHs, where stress energy within these BHs evolves with the expanding Universe in such a way that the BH mass changes as $M_{\text{BH}} \sim a^3$, independent of the accretion and merging of the galaxies. This description is very different from the one presented here (we consider the evolution of structures to follow the accretion and merging in the expanding Universe). However, it may be possible to link our study to the one of [38] by not using the standard link between BHs and halos (as we use here) $M_{\text{BH}} \sim M_{\text{halo}}^5$. We will leave such detailed comparisons for a future study.

Several limitations of the present approach relate to the calculation of the distribution of the small-scale structure. First of all, whereas the Press–Schechter formalism was the first and simplest method to analytically calculate the mass function, it has been demonstrated, in particular through the use of numerical cosmological simulations, that both the mass dependence and redshift evolution have somewhat different properties than those predicted by PS [39–41]. Secondly, whereas here, we simplify the full mass function as a simple power-law, in reality, one should integrate over the full distribution function.

In this discussion, it was assumed that all BHs follow the standard correlation with $M_{\mathrm{BH}} \sim \sigma_{\mathrm{halo}}^{5.1}$. It may be that the early Universe contained BHs that were significantly more massive [42–45], which, in particular, may change the details of the evolution of the Universe.

## 7. Conclusions

In order to ensure that effective field theories remain consistent, a relationship between the UV cut-off and the IR physics has been proposed [12] that suggests a relationship between Dark Energy and black holes. In order to test this connection, we present a time-dependent calculation that includes the evolution of all structure formation (which links to the evolving masses of BHs) in the expanding Universe. By comparison with cosmological SN1a data, we find that the simplest models of [12,15], where we extrapolated the observed BH–galaxy masses to be valid at all masses, are not in agreement with the expansion history as measured through SN1a. Instead, we find that another simple model for which the energy term is $\Delta E \sim M^{\beta+1}$ is in fairly good agreement with the SN1a data using $\beta \approx 0.2$. The limitations of the description presented above, which are dominated by the assumption that the mass spectrum of halos can be simplified by a single spectral index, implies that we cannot exclude the possibility that the accelerated expansion may be driven by an effect driven by the velocity dispersions of galaxies [18,19].

**Author Contributions:** Conceptualization, H.C. and S.H.H.; Methodology, H.C., B.T. and S.H.H.; Software, B.T.; Formal analysis, H.C., B.T. and S.H.H.; Investigation, B.T.; Writing—original draft, S.H.H.; Writing—review & editing, H.C. and B.T. All authors have read and agreed to the published version of the manuscript.

**Funding:** This research received no external funding.

**Data Availability Statement:** Data are contained within the article.

**Acknowledgments:** It is a pleasure thanking Zhen Li for interesting discussions in the early phase of this project. We thank the anonymous referees for excellent suggestions that improved the paper.

**Conflicts of Interest:** The authors declare no conflicts of interest.

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
