# Peer review of "Cosmological Test of an Ultraviolet Origin of Dark Energy"

_universe, doi:10.3390/universe10050193_

Round 1
Reviewer 1 Report
Comments and Suggestions for Authors
See report.

Reviewer 2 Report
Comments and Suggestions for Authors
Some references should be more transparent, like on page 1. sentence starts with "To address this issue, (12) proposed a relationship..." should be changed into "To address this issue, Cohen et al. (12) proposed a relatioship..."
Comments on the Quality of English Languagesee above!
Author Response
REFEREE
Some references should be more transparent, like on page 1. sentence starts with "To address this issue, (12) proposed a relationship..." should be changed into "To address this issue, Cohen et al. (12) proposed a relatioship..."
ANSWER
Thanks. We have changed this and similar references.
Reviewer 3 Report
Comments and Suggestions for Authors
Dear Editor,
I have reviewed for Universe the paper #2884115 titled "Cosmological test of an ultraviolet origin of Dark Energy" by Christiansen, Takacs and Hansen.
The authors presents a cosmological model where the Friedmann equation is modified to include an energy term related to the abundance of DM halos, in turn possibly tracing back its origin to surface area entropy of supermassive BHs.
I am afraid I cannot recommend the paper for publications in Universe, since I have strong concerns both from a theoretical point of view, and in relation to the comparison with data, as detailed below:
*) Section 3. The physical rationale underlying the proposed Eq. 2 is unclear. The Friedmann equation is not something coming out of the blue: it is a solution of general relativity for a homogeneous and isotropic Universe (or at least a patch of it). From this perspective, it seems weird to couple the Friedann equation to the mass of dark matter halos, which live and evolve on much smaller scales.
I mean, on the scales of virialized dark matter halos hosting supermassive BHs (e.g., 10^13 Msun) the Universe in not at all homogeneous and isotropic. Alternatively, do the authors mean that BHs on small scales can have a backreaction effects on cosmological scales? How can it occur in physical terms without completely altering the geometry (e.g., with a nonlocal coupling with the gravitational metric)? I cannot see any realistic physical process that can confidently establish such a connection in terms of a modified Friedmann equation, unless invoking some hand-waving quantum-gravity argument that honestly is highly exotic and poorly defined.
*) Section 3. Eq. 5 holds for a scale-invariant power spectrum, but the value n=1 adopted by the authors is odd. The standard cold DM power spectrum can be approximated by a scale-dependent n, that goes from n ~ -2.5 for small halos (hosting dwarf galaxies) out to n ~ -1 for the largest ones (hosting galaxy clusters). Actually n ~ +1 describes the initial Harrison-Zel'dovich matter spectrum, but this cannot be used for structure formation because many processes modify it before recombination that give the aforementioned range for n (in fact, the power spectrum at decoupling is usually rendered in terms of a transfer function that multiplies the initial spectrum).
*) Section 3. The Press & Schechter mass function is known to poorly describe the abundance of halos in the Universe. Admittedly it was one of the first and simplest model to be developed, but with the advent of N-body simulations people started to realize that it performs very poorly both in terms of mass dependence and redshift evolution, especially around the characteristic mass M^star used by the authors. Many detailed fits to the N-body outcomes are available in the literature (e.g., Sheth+02,Tinker+08, Shirasaki+21, and many others).
*) Section 4. There is a vicious circle here. To include an additional energy term into the Friedmann equation comes at a price. There is a new energy density component in the Universe which will satisfy a mass-energy conservation equation different from that of CDM. Even if such component do not couple directly to CDM, it would do so indirectly via gravity, and this will imply that the evolution of CDM perturbations and the shape of the mass function would be ensuingly affected to some extent.
*) Section 5. The authors show that with two free parameters their model can fit the SN data out to z=2. Unfortunately, this is a very poor test of a cosmological model. There are plenty of models that can be constructed to fit SN data with 2 or more free parameters; still they dramatically fail when compared jointly to the main cosmological observables, like BAO or CMB spectrum. To robustly advocate that a cosmological model works, a minimal requirement is to fit jointly SN data, volume-average baryon acoustic oscillations, and the position of the first CMB peak. It is not sufficient to say that the expansion history is similar to that of LCDM to fit BAO and CMB, because these are tests with very stringent observational constraints: even small difference in the expansion history and geometrical distances may be critical.
*) Section 6. The reasoning presented here is, astrophysically speaking, very weird. First, the relation between BH masses and halo masses or velocity dispersion has a very well known astrophysical origin, connected to the energy feedback from supermassive BH during the galaxy formation process; so nothing to do with cosmology at large.
Second, this relation cannot be applied to all halos present in the mass function. (Super)massive BHs are ubiquitous in massive quiescent galaxies, and their properties correlate with those of the galaxies (and possibly with those of the host halo) only there. Such massive galactic halos actually constitute only a small fraction of the halos present in the mass function; therefore if one traces back the new energy term to supermassive BHs, to use the *overall* halo mass function is wrong.
Third, the bestfit model by the authors require that the energy density added to the Friedmann equation scales monotonically with the redshift (or scale factor). But this is at variance with the energy density in supermassive BHs, which instead increases with redshift, has a peak at around redshift 3 and then declines at higher z (e.g., it roughly follows the cosmic SFR density of galaxies).
Last but not the least, it should be remarked that the number density in stellar mass BHs (so the ones associated to the death of massive stars in galaxies) largely overwhelms, by 6-8 orders of magnitude at z=0, that in supermassive BHs at the center of galaxies (so the ones for which BH mass correlates with host halo mass). Thus calling into the game the latter and not considering the former appear specious.
Comments on the Quality of English LanguageOverall good quality of English phrasing, check for minor typos.
Round 2
Reviewer 1 Report
Comments and Suggestions for Authors
See pdf.

Reviewer 3 Report
Comments and Suggestions for Authors
Dear Editor,
I thank the authors for having taken my comments and suggestions into account. Despite the strong assumptions and large limitations of this work, I think that the authors have now fairly stated these in the revised text. Therefore, I can recommend the paper for publication.